# Impact of Reactive Sulfur Species on *Entamoeba histolytica*: Modulating Viability, Motility, and Biofilm Degradation Capacity

**DOI:** 10.3390/antiox13020245

**Published:** 2024-02-19

**Authors:** Jun Ye, Talal Salti, Eva Zanditenas, Meirav Trebicz-Geffen, Moran Benhar, Serge Ankri

**Affiliations:** 1Department of Molecular Microbiology, Ruth and Bruce Rappaport Faculty of Medicine, Technion-Israel Institute of Technology, Haifa 3525433, Israelmeiravg@technion.ac.il (M.T.-G.); 2Department of Biochemistry, Ruth and Bruce Rappaport Faculty of Medicine, Technion-Israel Institute of Technology, Haifa 3525433, Israel

**Keywords:** reactive sulfur species, hydrogen sulfide, cysteine persulfide, S-sulfuration, *Entamoeba histolytica*

## Abstract

Reactive sulfur species (RSS) like hydrogen sulfide (H_2_S) and cysteine persulfide (Cys-SSH) emerged as key signaling molecules with diverse physiological roles in the body, depending on their concentration and the cellular environment. While it is known that H_2_S and Cys-SSH are produced by both colonocytes and by the gut microbiota through sulfur metabolism, it remains unknown how these RSS affect amebiasis caused by *Entamoeba histolytica*, a parasitic protozoan that can be present in the human gastrointestinal tract. This study investigates H_2_S and Cys-SSH’s impact on *E. histolytica* physiology and explores potential therapeutic implications. Exposing trophozoites to the H_2_S donor, sodium sulfide (Na_2_S), or to Cys-SSH led to rapid cytotoxicity. A proteomic analysis of Cys-SSH-challenged trophozoites resulted in the identification of >500 S-sulfurated proteins, which are involved in diverse cellular processes. Functional assessments revealed inhibited protein synthesis, altered cytoskeletal dynamics, and reduced motility in trophozoites treated with Cys-SSH. Notably, cysteine proteases (CPs) were significantly inhibited by S-sulfuration, affecting their bacterial biofilm degradation capacity. Immunofluorescence microscopy confirmed alterations in actin dynamics, corroborating the proteomic findings. Thus, our study reveals how RSS perturbs critical cellular functions in *E. histolytica*, potentially influencing its pathogenicity and interactions within the gut microbiota. Understanding these molecular mechanisms offers novel insights into amebiasis pathogenesis and unveils potential therapeutic avenues targeting RSS-mediated modifications in parasitic infections.

## 1. Introduction

RSS collectively refers to a variety of sulfur-containing chemical species that exhibit reactivity and are involved in various biological and chemical processes. H_2_S and Cys-SSH are often considered two of the most important RSSs due to their significant roles in biological systems [1]. H_2_S is a gas that was once considered a toxic byproduct of metabolic processes in living organisms. However, recent research showed that H_2_S plays crucial roles in various physiological and pathophysiological processes depending on its concentration and location within a cell or organism [2,3]. H_2_S is produced by several enzymes, including cystathionine gamma-lyase (CSE) [4], cystathionine beta-synthase (CBS), and 3-mercaptopyruvate sulfurtransferase (3-MST), which are expressed in different organs and tissues in vivo [2]. Moreover, it was recently suggested that H_2_S (together with H_2_O_2_) can also be produced by a fourth endogenous enzyme, selenium-binding protein 1 (SELENBP1). This enzyme plays a crucial role in the colon, producing H_2_S through the oxidation of methanethiol, a product of methionine metabolism in gut bacteria [5]. One of the most important aspects of H_2_S function is its ability influence diverse signaling pathways (e.g., NF-κB, MAPK, HIF-1α, NRF-2) and cell survival [6,7]. The gut microbiota are a complex ecosystem that is composed of trillions of microorganisms, including bacteria, archaea, fungi, and viruses. These microorganisms interact with each other and with the host and play important roles in digestion, metabolism, and immune function [8]. One of the metabolic processes that the gut microbiota are involved in is sulfur metabolism, which leads to the production of H_2_S [9]. The gut microbiota produces H_2_S through two pathways: sulfate reduction by bacteria like *Desulfovibrio* and thiosulfate reduction by bacteria such as *Escherichia coli* and *Salmonella* [10,11]. The production of H_2_S by the gut microbiota are regulated by various factors, including diet, host genetics, and microbial interactions. The gut microbiota notably influences H_2_S production and maintenance in the body. Colonic bacteria play a significant role in generating substantial amounts of H_2_S, resulting in its commonly observed presence in the gut at millimolar concentrations [12,13,14]. Additionally, research indicates that modifications in the composition and function of the gut microbiota can affect H_2_S production and metabolism, thereby holding significant implications for human health [9,15]. Concerning Cys-SSH, its formation in colonocytes is catalyzed by CSE and by CBS using cystine as a substrate [4,16]. Additionally, cysteinyl–tRNA synthetases were identified as enzymes capable of generating Cys-SSH using cysteine as a substrate [17]. Similar to H_2_S, gut bacteria also contribute to the production of Cys-SSH during sulfur metabolism in the gastrointestinal tract [18].

Amebiasis is a parasitic infection caused by the protozoan parasite *Entamoeba histolytica*. It is a common cause of diarrheal illness worldwide, particularly in developing countries with poor sanitation and hygiene. The infection is typically acquired through ingestion of contaminated food or water and can induce various symptoms such as diarrhea, abdominal pain, and fever. In severe cases, the infection can lead to the development of abscesses in the liver or other organs, which can be life-threatening [19]. Treatment typically involves a course of antibiotics, with metronidazole being the standard therapy for treating adults and children with invasive amoebiasis [20,21].

The presence of RSS in the gut [18] prompts inquiry into their potential role in regulating amebiasis. Previous studies established that *E. histolytica* can generate H_2_S from cysteine through the actions of methionine gamma-lyase 1&2 [22,23]. Moreover, apart from producing H_2_S internally, the parasite is likely exposed to H_2_S generated by the gut microbiota. Recent research suggests that the cellular effects of H_2_S involve, in part, protein S-sulfuration (also referred to as S-persulfidation), which can modify protein conformation, function, and activity [24]. The presence of Cys-SSH in the gut also suggests that S-sulfuration may occur directly in the parasite. Despite these intriguing findings, our understanding of the influence of RSS, such as H_2_S and Cys-SSH, on the growth and physiology of *E. histolytica* remains limited. To gain a deeper understanding of the role of RSS in *E. histolytica*, we conducted in parallel a physiological study and a proteomic analysis of S-sulfurated proteins. This study provides valuable insights into the regulation of H_2_S and Cys-SSH in the parasite, and it offers promising avenues for further research into the role of these RSS in the pathogenesis of amebiasis and their potential as a therapeutic target for this disease.

## 2. Materials and Methods

*E. histolytica* culture

*E. histolytica* strain HM-1:IMSS was generously provided by Prof. Samudrala Gourinath at Jawaharlal Nehru University, New Delhi, India. *E. histolytica* was maintained at 37 °C in 13 × 100 mm screw-capped Pyrex glass tubes containing Diamond’s TYI-S-33 medium until reaching the exponential growth phase. Trophozoites were harvested from the culture tubes by tapping them, followed by centrifugation, as described in previous protocols [25].

Determination of Lethal Dose, 50% (LD_50_) of Na_2_S or Cys-SSH for *E. histolytica*

Trophozoites (1 × 10^6^/mL) were treated with 2.0–3.5 mM Na_2_S (Merck, Rehovot, Israel) or 0.2–1.2 mM Cys-SSH. A modified procedure for Cys-SSH production and concentration determination, as described in [26,27,28], involved dissolving 20 mM cystine (Thermo Fisher Scientific (Heysham), Lancashire, UK) in degassed distilled water, adjusting the pH to 8.8, transferring 5 mL to a 15 mL tube, adding 15 mg Na_2_S, and subsequently, transferring the mixture to an Eppendorf tube for a 30 min incubation at 37 °C. This was followed by measuring the concentration of Cys-SSH using the cold cyanolysis method [29,30]. It is expected that Cys-SSH is the major compound in the solution, but the presence of unreacted Na_2_S and cystine cannot be ruled out [28,30]. Freshly prepared Cys-SSH solutions were systematically utilized in this study. Na_2_S or Cys-SSH treatments of the trophozoites were carried out in phosphate-buffered saline (PBS) containing glucose 0.1% (PBS–glucose) at different times at 37 °C. PBS–glucose was selected instead of Diamond’s TYI-S-33 medium due to the latter containing cysteine at millimolar concentrations [25]. This high concentration of cysteine can potentially interfere with Na_2_S or Cys-SSH treatments, diminishing their anti-amebic activity (Appendix A). The number of viable trophozoites was counted by using the eosin dye exclusion method [31].

Detection of intracellular concentration of H_2_S in trophozoites by Washington State Probe-5 (WSP-5) coupled to flow cytometry

To assess intracellular H_2_S levels, 1 × 10^6^ trophozoites were harvested and treated with or without 2.65 mM Na_2_S for 5–30 min in PBS–glucose. After treatment, the cells were then washed three times with serum-free TYI medium by centrifugation at 1900 rpm for 3 min. Subsequently, the suspended trophozoites were incubated with 1 mL of serum-free TYI medium containing 50 µM of WSP-5 (Cayman Chemical, Ann Arbor, MI, USA) for 15 min at 37 °C. WSP-5 is a fluorescent probe for H_2_S that emits fluorescence with excitation/emission maxima of 502/525 nm, respectively, upon reaction with H_2_S. After washing the cells with PBS three times, they were suspended in 1 mL of PBS and analyzed by flow cytometry. Flow cytometry was performed using BD LSRFortessa™ Cell Analyzer (Becton, Dickinson and Company, Franklin Lakes, NJ, USA), and data from 10,000 cells were collected under each condition.

Detection of intracellular concentration of persulfides and polysulfides in trophozoites by Sulfane Sulfur Probe 4 (SSP4) coupled to flow cytometry

To assess intracellular persulfides and polysulfides levels, 1 × 10^6^ trophozoites were harvested and treated with or without 0.8 mM Cys-SSH, 0.8 mM L-Cystine (Thermo Fisher Scientific (Heysham), Lancashire, UK), 0.8 mM L-Cysteine (Merck Millipore, Rosh-Ha’ayin, Israel) or with 0.8 mM Na_2_S (Merck, Rehovot, Israel) for 30 min in PBS–glucose. After treatment, the cells were then washed three times with serum-free TYI medium by centrifugation at 1900 rpm for 3 min. Subsequently, the suspended trophozoites were incubated with 1 mL of serum-free TYI medium containing 20 µM of SSP4 (Cayman Chemical, Ann Arbor, MI, USA) for 15 min at 37 °C. SSP4 is a fluorescent probe that emits fluorescence with excitation/emission maxima of 482/515 nm, respectively, upon reaction with sulfane sulfur. After washing the cells with PBS three times, they were suspended in 1 mL of PBS and analyzed by flow cytometry. Flow cytometry was performed using BD LSRFortessa™ Cell Analyzer (Becton, Dickinson and Company, Franklin Lakes, NJ, USA), and data from 10,000 cells were collected under each condition.

Biotin Thiol Assay (BTA)

To identify S-sulfurated proteins in the parasite exposed to Cys-SSH, we conducted a BTA coupled with LC-MS/MS analysis [32,33]. In the BTA assay, Cys-S-SH groups formed as a result of persulfidation reaction with biotin-conjugated maleimide. These Cys-S-S-biotin-tagged proteins are selectively captured by an avidin column. Upon elution using dithiothreitol (DTT), previously sulfhydrated proteins are specifically released. Proteins that have not undergone sulfhydration retain their Cys-SH groups, which react with biotin-conjugated maleimide to form Cys-biotin. These unmodified proteins cannot be eluted by DTT from the avidin column, allowing for the precise isolation of sulfhydrated proteins. The assay was performed as follows: 1 × 10^7^ trophozoites were harvested and treated with or without 0.8 mM Cys-SSH for 30 min in PBS–glucose. Cells underwent lysis using RIPA buffer (containing 150 mM NaCl, 1% NP-40, 0.5% sodium deoxycholate, 50 mM Tris-HCl (pH 7.4), and 0.1% SDS in PBS) on ice for 10 min. Afterward, lysates were clarified by centrifugation at 4 °C, 13,000 rpm, for 20 min (using the Eppendorf Centrifuge 5810 R), and protein concentrations were determined using the bicinchoninic acid (BCA) assay (BioRad, Rishon Le Zion, Israel). Equal amounts (2 mg) of proteins were then mixed with 0.1 μM biotin–polyethylene glycol (PEG) 2-maleimide (Thermo Fisher Scientific, Waltham, MA, USA) in 2.5 mL RIPA buffer and incubated at 37 °C for 30 min with occasional gentle mixing. The resulting mixture was subsequently precipitated by cold acetone. After centrifugation, the precipitated protein pellets were washed four times with 70% cold acetone, followed by suspension in a buffer (0.1% SDS, 150 mM NaCl, 1 mM EDTA, 0.5% Triton X-100, and 50 mM Tris-HCl, pH 7.5). This suspension was mixed with 50 μL Streptavidin agarose resin (Thermo Fisher Scientific, Waltham, MA, USA) and rotated overnight at 4 °C. Post-incubation, the beads were washed five times with wash buffer 1 (0.5% Triton X-100, 150 mM NaCl, 50 mM Tris-HCl, pH 7.5), followed by seven washes with wash buffer 2 (0.5% Triton X-100, 600 mM NaCl, 50 mM Tris-HCl, pH 7.5). Resin with bound proteins underwent incubation with 500 μL of elution buffer, with or without 20 mM DTT, for 30 min at 25 °C. The eluted proteins were concentrated to a final volume of 25–35 μL using an Amicon Ultracel 10 K (Merck Millipore, Rosh-Ha’ayin, Israel) column. Following the addition of 5× sample buffer (250 mM Tris (pH 6.8), 12% SDS, 50% glycerol, 1 mg/mL bromophenol blue) and β-mercaptoethanol, the concentrated proteins were boiled for 10 min at 100 °C. Subsequently, the resulting samples were subjected to gel electrophoresis (10% sodium dodecyl sulfate polyacrylamide gel electrophoresis (SDS-PAGE)), followed by mass spectrometry (MS) analysis.

In gel proteolysis and MS analysis

Gel-based proteolysis was conducted following a previously outlined procedure [34]. The proteins within the gel were first reduced with 3 mM DTT (60 °C for 30 min), followed by modification with 10 mM iodoacetamide in 100 mM ammonium bicarbonate (carried out in a light-protected environment at room temperature for 30 min). Subsequently, the proteins were enzymatically digested in a solution comprising 10% acetonitrile and 10 mM ammonium bicarbonate with modified trypsin (Promega, Beit Haemek, Israel) at an enzyme-to-substrate ratio of 1:10, overnight at 37 °C. A secondary digestion with Trypsin was then carried out for 4 h at 37 °C. The resultant tryptic peptides were desalted using C18 tips (home-made stage tips), dried, and re-suspended in 0.1% formic acid. These peptides were further separated by reverse-phase chromatography on a 0.075 × 300 mm fused silica capillary (J&W, Agilent Technologies, Santa Clara, CA, USA) packed with Reprosil reversed-phase material (Dr Maisch GmbH, Ammerbuch, Germany). Elution of peptides was achieved through a linear gradient of 5% to 28% acetonitrile with 0.1% formic acid in water over 60 min, followed by a 15 min gradient of 28% to 95% acetonitrile with 0.1% formic acid in water, and finally, 15 min at 95% acetonitrile with 0.1% formic acid in water, all at a flow rate of 0.15 μL/min. Mass spectrometry analysis was performed using a QExactive plus mass spectrometer (Thermo Fisher Scientific, Waltham, MA, USA) in positive mode, employing repetitive full MS scans followed by High Collision Dissociation (HCD) of the 10 most dominant ions selected from the initial MS scan.

The MaxQuant software 2.1.1.0 [35] was employed to analyze the mass spectrometry data, facilitating peak picking and identification through the Andromeda search engine. The search was conducted against the *Entamoeba histolytica* section of the Uniprot database, with a mass tolerance of 4.5 ppm for precursor masses and 20 ppm for fragment ions. Variable modifications, including oxidation on methionine, protein N-terminus acetylation, and biotin on lysine, were accepted, while carbamidomethyl on cysteine was considered static. Peptides with a minimum length of seven amino acids and up to two miscleavages were allowed. Label-free analysis was performed using the same software for data quantification. Peptide- and protein-level false discovery rates (FDRs) were filtered to 1% using the target-decoy strategy. Protein tables underwent filtration to remove identifications from the reverse database and common contaminants. Statistical analysis of identification and quantification results was conducted using Perseus software(Perseus 1.6.7) (Mathias Mann’s group). Proteins were deemed significantly altered if their abundance exhibited a 2-fold change or more in the sample (WT + Cys-SSH) compared to the control (WT), with a *p*-value < 0.05 and Razor + unique peptides > 1.

PANTHER Classification System

The study utilized PANTHER Version 18.0 (http://pantherdb.org/; [36] accessed on 20 October 2023) for data analysis. Sulfhydrated proteins were categorized using the “protein class” ontology setting. The statistical overrepresentation test was conducted with default setting, utilizing the annotation data set corresponding to PANTHER protein class and selecting the FDR correction for multiple testing options.

Assessment of protein synthesis through surface sensing of translation (SUnSET)

Protein synthesis in trophozoites exposed to Cys-SSH (0.8 mM for 30 min) and untreated trophozoites (control) in PBS–glucose was assessed using SUnSET [37,38]. Briefly, treated trophozoites (2 × 10^6^) were incubated with 10 μg/mL puromycin (Merck, Rehovot, Israel), a structural analog of tyrosyltRNA, for 20 min at 37 °C. Following this, the trophozoites were lysed using 1% Igepal (Merck, Rehovot, Israel) in PBS. Whole proteins were separated on a 10% SDS-PAGE in SDS-PAGE running buffer and subsequently electrotransferred to a nitrocellulose membrane in protein transfer buffer. Loading equivalency was determined by Ponceau-S (Merck, Rehovot, Israel) staining of the membrane before immunostaining. Puromycin incorporation was detected by immunoblotting using a 1:1000 dilution of monoclonal puromycin antibody (12D10 clone, Merck Millipore, Rosh-Ha’ayin, Israel). Following incubation with the primary antibody, the blots were treated with a 1:5000 dilution of secondary antibody (Jackson ImmunoResearch, West Grove, PA, USA) for 2 h at room temperature, then developed using enhanced chemiluminescence (WesternBright^TM^ ECL, Advansta, CA, USA) and photographed with Fusion FX7 Edge Spectra. Protein synthesis quantification was determined from the intensity of the immunoreactive blots (densitometry) using Fiji software version 1.54f [39].

Immunofluorescence microscopy

A total of 1 × 10^6^
*E. histolytica* trophozoites were harvested and treated with or without 0.8 mM Cys-SSH for 30 min in PBS–glucose. Then, trophozoites (1.5 × 10^5^ trophozoites/mL) were suspended in TYI medium without serum at 37 °C and transferred onto glass coverslips cleaned with acetone, positioned at the bottom of each well of a 24-well plate. Following this, trophozoites were allowed to adhere to the coverslip surface during a 1 h incubation at 37 °C. Subsequently, the attached trophozoites were fixed with prewarmed 37 °C 4% paraformaldehyde (PFA, Electron Microscopy Sciences, Hatfield, PA, USA) for 30 min at room temperature. After fixation, trophozoites were permeabilized with 0.1% Triton X-100/PBS for 1 min at room temperature. The coverslips were washed three times with PBS and quenched with PBS containing 50 mM NH_4_Cl for 30 min at room temperature. Following this, the coverslips were blocked with 1% bovine serum albumin (BSA, MP Biomedicals, Solon, OH, USA) in PBS (BSA/PBS) for 1 h at room temperature. The samples were then incubated overnight with a 1:1000 dilution of monoclonal actin antibody (clone C4, MP Biomedicals, Solon, OH, USA), known to effectively detect *E. histolytica* actin [40]. The next day, the samples underwent PBS and 1% BSA/PBS washes, followed by a 4 h incubation at 4 °C with a 1:250 dilution of Alexa Fluor 488 (Jackson ImmunoResearch, West Grove, PA, USA) and a 1:1000 dilution of 4’,6-diamidino-2-phenylindole (DAPI; MP Biomedicals, Solon, OH, USA). After incubation, coverslips were washed in PBS and then incubated overnight at 4 °C with a 1:500 dilution of Phalloidin-iFluor 594 Reagent (Abcam, Cambridge, UK). Following additional washing steps, coverslips were mounted onto microscope slides using Fluoromount G (SouthernBiotech, Birmingham, AL, USA). The specimens were examined under a confocal immunofluorescence microscope (ZEISS-LSM700 Meta Laser Scanning System) equipped with a 63× oil immersion objective. Fluorescent quantification of control trophozoites and those treated with Cys-SSH was conducted using Fiji software version 1.54f [39].

Determination of *E. histolytica* motility

Using the Costar Transwell system (8 μm pore size polycarbonate membrane, 6.5 mm diameter, Corning Inc., New York, NY, USA) as outlined in [41], trophozoite motility was evaluated. Briefly, 1 × 10^6^
*E. histolytica* trophozoites were collected and treated with or without 0.8 mM Cys-SSH for 30 min in PBS–glucose. The trophozoites were then suspended in serum-free TYI medium. A 500 μL aliquot of the suspension (5 × 10^5^ trophozoites/mL) was loaded into a transwell insert, which was placed in each well of a 24-well culture plate containing 500 μL serum-free TYI medium. The 24-well culture plate, along with the loaded inserts, was sealed in anaerobic bags (Mitsubishi Gas Chemical Company, Inc., Tokyo, Japan) and incubated for 3 h at 37 °C. Upon completion of the incubation, the inserts and culture medium were removed from each well, and trophozoite migration was assessed by enumerating the number of trophozoites adhered to the bottom of each well.

CPs activity assay

*E. histolytica* trophozoites were harvested and treated with different conditions separately: control trophozoites (WT), Cys-SSH-treated trophozoites (0.8 mM Cys-SSH in PBS–glucose for 30 min), and E64D-treated trophozoites (10 µM E64D in Diamond’s TYI-S-33 medium for overnight). CP activity was evaluated in total lysates of differently treated *E. histolytica* trophozoites (1 × 10^6^) that were lysed in 1 mL of 1% Nonidet P-40 (NP-40) solution prepared in deuterium-depleted water (DDW). The assessment of CP activity was conducted by monitoring the degradation of the chromophoric substrate benzyloxycarbonyl-L-arginyl-L-arginine-p-nitroanilide (Z-Arg-Arg-pNA; Bachem, Bubendorf, Switzerland) [42]. One unit of CP activity was defined as the enzyme quantity capable of digesting one micromole of Z-Arg-Arg-pNA per minute per milligram of protein.

Gelatin gel method

*E. histolytica* trophozoites were harvested and treated with different conditions separately: control trophozoites (WT), Cys-SSH-treated trophozoites (0.8 mM Cys-SSH in PBS–glucose for 30 min), and E64D-treated trophozoites (10 µM E64D in Diamond’s TYI-S-33 medium for overnight). To assess the CP activity, 1 × 10^6^ *E. histolytica*-treated trophozoites were lysed using 0.5% NP-40 for 10 min on ice. Following this, lysates were clarified by centrifugation at 4 °C, 12,000 rpm, 1 min (Eppendorf Centrifuge 5810 R), and the protein concentrations were assessed using the BCA assay (BioRad, Rishon Le Zion, Israel). SDS-PAGE contained 1% gelatin (Merck, Rehovot, Israel) and was prepared previously. A total of 40 µg of protein in 1× sample buffer without DTT were loaded onto the gel and subjected to electrophoresis at a constant voltage of 150 volts. Subsequently, the gel was washed three times for 10 min each using a washing solution comprising 2.5% Triton X-100. The gel was then placed in a developing buffer, which consisted of 100 mM sodium acetate with a pH of 4.2 and 1% Triton X-100, and incubated at 37 °C for 30 min. Finally, the gel was treated with Coomassie Blue staining solution to visualize the protein bands.

*B. subtilis* biofilm formation

Biofilm formation of *B. subtilis* was initiated using *B. subtilis* GFP-expressing strain NCIB3610, specifically the *amyE::Phyperspank-gfp* variant [43]. A single colony was selected from lysogeny broth (LB) plates and cultivated until mid-logarithmic phase in a 3 mL LB culture, with agitation at 37 °C for 4 h at 200 rpm using a New Brunswick scientific, Innova 4300 shaker. To prepare the biofilm, we adapted a method from Xiaoling Wang et al. [44]. Cells from the mid-logarithmic phase were diluted 1:10 into a serum-free TYI medium and incubated overnight at 30 °C without agitation. Growth occurred in 24-well plates, each well containing 1 mL of serum-free TYI agar.

*B.subtilis* biofilm degradation assay by *E. histolytica* trophozoites

The procedure used to determine the degradation of *B.subtilis* by *E. histolytica* trophozoites is described in [45]. *E. histolytica* trophozoites were harvested and treated with different conditions separately: control trophozoites (WT), Cys-SSH-treated trophozoites (0.8 mM Cys-SSH in PBS–glucose for 30 min), and E64D-treated trophozoites (10 µM E64D in Diamond’s TYI-S-33 medium for overnight). Treated trophozoites (1 × 10^6^) were placed onto *B. subtilis* biofilm and incubated at 37 °C for 3 h without agitation. To measure the degree of biofilm degradation, photographs of the biofilms were captured using an Olympus stereoscope, and the GFP signal intensity of each well was assessed relative to the control (biofilm incubated without trophozoites) using Fiji software version 1.54f [39]. The original images containing the GFP signal were converted into grayscale images, where white indicates a stronger GFP signal and black represents a weaker signal. Thus, a darker image indicates a greater degree of biofilm destruction and a weaker biofilm. The amount of black pixels in the image, which corresponds to the extent of biofilm degradation, was quantified and normalized relative to the control sample.

Detection of ROS in *E. histolytica* trophozoites

The trophozoites were treated with 0.8 mM Cys-SSH in PBS–glucose for 30 min or 2.5 mM H_2_O_2_ in PBS–glucose for 15 min at 37 °C. Trophozoites were washed once with serum-free TYI medium to eliminate the Cys-SSH or H_2_O_2_ and resuspended in a 24-well plate. Each well was treated with a fluorescent probe, 2′,7′-dichlorodihydrofluorescein diacetate (H2DCFDA, 10 μM, Merck, Rehovot, Israel), and incubated for 30 min at 37 °C in the absence of light, within TYI medium lacking serum. Following this, the trophozoites were fixed with 4% PFA for 30 min at room temperature in the absence of light. The trophozoites were analyzed using a fluorescence microscope, and the fluorescence intensity of the H2DCFDA probe was utilized to quantify the level of oxidation in each trophozoite. The quantification of ROS levels in trophozoite cells was carried out using Fiji software version 1.54f [39].

For detecting ROS level by flow cytometry, 1 × 10^6^ trophozoites were harvested and treated with or without 0.08 mM, 0.4 mM, or 0.8 mM Cys-SSH in PBS–glucose for 30 min. After treatment, the cells were then washed three times with serum-free TYI medium by centrifugation at 1900 rpm for 3 min. Subsequently, the suspended trophozoites were incubated with 1 mL of serum-free TYI medium containing 20 µM of H2DCFDA probe for 15 min at 37 °C. H2DCFDA probe is a fluorescent probe for ROS that emits fluorescence with excitation/emission maxima of 504/529 nm, respectively, upon reaction with ROS. After washing the cells with PBS three times, they were suspended in 1 mL of PBS and analyzed by flow cytometry. Flow cytometry was performed using BD LSRFortessa™ Cell Analyzer (Becton, Dickinson and Company, Franklin Lakes, NJ, USA), and data from 10,000 cells were collected for each condition.

Statistical analysis

Statistical analysis and graphical representations were conducted utilizing Prism 9 (Graphpad Software Inc., San Diego, CA, USA). The data are presented as mean ± standard error of the mean (SEM) from 2 to 3 biological replicates. Unless otherwise specified, significance was assessed using one-way ANOVA for multiple comparisons and the Mann–Whitney test for two groups of comparisons. Normality was tested by Shapiro–Wilk test.

## 3. Results

### 3.1. Exogenous H_2_S and of Cys-SSH Induce Rapid Cytotoxicity in E. histolytica Trophozoites

H_2_S is produced by colonic tissue through the activity of transsulfuration enzymes (CBS, CSE) and is also derived from bacteria, leading to concentrations of up to 3.4 mmol/L (as measured in human stools) [14]. The presence of H_2_S in the gut and its emerging role in inhibiting the release of invasive pathobionts [46] suggest its potential relevance to the biology of *E. histolytica*. Subsequently, we carried out experiments to explore the impact of H_2_S on the viability of *E. histolytica*. To achieve this, we utilized Na_2_S, which rapidly releases H_2_S in aqueous solution [47]. After subjecting *E. histolytica* trophozoites to varying concentrations of Na_2_S for 30 min, we established that the LD_50_ of Na_2_S corresponds to 2.65 mM (Figure 1a). At 2.65 mM Na_2_S, we observed that 75% of the trophozoites were killed after one hour of exposure (Figure 1b). In order to monitor the H_2_S delivery into the parasite, we used fluorescent probe WSP-5 [48]. At a concentration of 2.65 mM Na_2_S, we observed a 1.4-fold increase in intracellular WSP-5 fluorescence compared to the untreated control trophozoites after 10 min of exposure. These results indicate that the addition of Na_2_S to the trophozoites rapidly leads to the accumulation of intracellular H_2_S (Figure 1c). S-sulfuration represents a recently identified post-translational modification promoted by H_2_S. This process chemically alters cysteine residues within the target protein, forming persulfides [49]. The cytotoxic impact of H_2_S on the parasite implies that essential biological functions within the parasite are disrupted by H_2_S. This suggests that the cytotoxic mechanism may be linked to the S-sulfuration of vital proteins, underscoring the potential significance of this post-translational modification in mediating H_2_S-induced toxicity. Moreover, the presence of Cys-SSH in the gut [18] suggests that the direct S-sulfuration of proteins facilitated by this trans-persulfidating agent [50] can also occur in the host. To investigate this hypothesis, trophozoites were subjected to Cys-SSH exposure, revealing that the LD_50_ of Cys-SSH in *E. histolytica* trophozoites is 0.8 mM after a 30 min treatment (Figure 2a). To monitor the Cys-SSH delivery into the parasite, we utilized the fluorescent reagent SSP4, commonly used for the detection of sulfane sulfurs [51]. While Cys-SSH led to the identification of sulfane sulfurs in the parasite, none of the components used to prepare Cys-SSH, namely cystine and Na_2_S [29], led to the detection of sulfane sulfurs (Appendix A).

### 3.2. Proteomic Profiling of S-Sulfurated Proteins in Parasites Exposed to Cys-SSH

We proceeded to detect S-sulfurated proteins by combining the BTA with MS-based proteomics [52] (Figure 2b). Using this approach, we identified 546 S-sulfurated proteins (*p*-value < 0.05, fold change > 2) in trophozoites exposed to Cys-SSH. These 546 S-sulfurated proteins were then classified using the PANTHER classification system [36] (Figure 2c). The seven most abundant S-sulfurated protein families belong to metabolite interconversion enzyme (exemplified by peroxiredoxin [EHI_201250]), protein-binding activity modulator (exemplified by Rho family GTPase [EHI_197840]), translational protein (exemplified by elongation factor 2 [EHI_189490]), protein-modifying enzyme (exemplified by cysteine proteinase 1 [EHI_074180]), transporter (exemplified by importin beta family protein 3 [EHI_098340]), membrane traffic protein (exemplified by adhesin 112 [EHI_181220]), and cytoskeletal protein (exemplified by actin [EHI_107290]). According to the PANTHER statistical overrepresentation test, which involves comparing a user input list of genes against a reference list containing all genes, a significant enrichment (with both *p*-value and q-value < 0.05) was observed for proteins annotated as chaperonin (exemplified by chaperonin containing TCP-1 subunit zeta [EHI_125800]), non-motor actin-binding protein (exemplified by F-actin-capping protein subunit beta [EHI_005020]), ribosomal protein (exemplified by 60S ribosomal protein L27 [EHI_183480]), vesicle coat protein (exemplified by Transmembrane protein tmp21 [EHI_058320]), actin or actin-binding cytoskeletal protein (PC00041) (exemplified by actin [EHI_107290]), translational protein (exemplified by elongation factor 2 [EHI_189490]), and cytoskeletal protein (exemplified by Tubulin alpha chain [EHI_005950]) (Figure 2d).

### 3.3. S-Sulfuration of Central Proteins in E. histolytica Impairs Their Function

The identification of ribosomal and translational proteins among the sulfhydrated proteins in this study (Figure 2c,d) suggests that this post-translational modification regulates translation in the parasite. To test this hypothesis, we employed the SUnSET method [53] to monitor changes in protein synthesis in both control trophozoites and trophozoites exposed to Cys-SSH. Our findings revealed a significant impairment of translation in the parasite upon exposure to Cys-SSH (Figure 3). These results suggest that the inhibition of protein synthesis in the parasite through S-sulfuration contributes to the cytotoxic activity of H_2_S.

The significant enrichment of actin or actin-binding cytoskeletal proteins among sulfhydrated proteins (Figure 2c,d) suggests that this post-translational modification regulates cytoskeletal activity in the parasite. Filamentous actin (F-actin) is a vital component of the parasite’s cytoskeleton [54]. Therefore, comparing the level of F-actin to the overall actin level in the parasite can serve as an indicator of its cytoskeletal activity [55]. Our findings revealed that the level of F-actin formed in trophozoites exposed to Cys-SSH is significantly lower than in untreated trophozoites (Figure 4). These results strongly suggest that the inhibition of F-actin formation (or induction of actin disassembly) in the parasite through S-sulfuration might influence essential cytoskeletal functions such as motility. To test this hypothesis, we evaluated the motility of trophozoites exposed to Cys-SSH using a transwell migration assay. We observed that the motility of trophozoites treated with Cys-SSH was reduced by 76% compared with control trophozoites (Figure 5).

*E. histolytica* CPs (EhCPs) serve as crucial virulence factors, enabling the parasite to infiltrate host tissues, evade the immune system, and acquire essential nutrients for survival during amoebiasis [56]. The presence of EhCPs such as EhCP1 among the S-sulfurated proteins suggests that sulfhydration may regulate their activity. To test this hypothesis, we measured EhCP activity by monitoring the cleavage of a specific CP substrate, z-Arg-Arg-pNA, in control trophozoites, trophozoites exposed to E64D (a cell-permeable CP inhibitor) [57], and trophozoites exposed to Cys-SSH (Figure 6c). As previously reported, EhCP activity was almost entirely inhibited in parasites cultivated in the presence of E64D [45], and it was significantly reduced in parasites exposed to Cys-SSH. We also employed gelatin zymography to assess CP activity in parasites exposed or not to Cys-SSH. Control trophozoites exhibited robust proteolytic activity toward gelatin, which was inhibited in trophozoites cultivated in the presence of E64D and in trophozoites exposed to Cys-SSH (Figure 6a,b). These results collectively indicate that Cys-SSH inhibits EhCP activity.

EhCPs are involved in the degradation of bacterial biofilms by the parasite [45]. Therefore, we tested the ability of Cys-SSH-treated trophozoites to degrade *B. subtilis* biofilm. We observed that Cys-SSH-treated trophozoites are impaired in their ability to degrade biofilm compared to untreated trophozoites (Figure 7). This, overall, suggests that the S-sulfuration of EhCPs may have consequences for the interaction between the parasite and bacterial biofilm present in the gut [45,58].

Many proteins related to redox regulation were identified as S-sulfuration targets (Figure 2c,d), indicating the potential involvement of this post-translational modification in the redox-sensitive pathways. To test this hypothesis, we quantified ROS levels in trophozoites treated with Cys-SSH compared to control trophozoites. Our findings demonstrated a significant increase in ROS levels in trophozoites treated with Cys-SSH, comparable to the ROS levels observed after H_2_O_2_ treatment (Figure 8a,b). We also assessed the ROS levels in trophozoites treated with varying concentrations of Cys-SSH using flow cytometry. Our results demonstrated a notable elevation in ROS levels with 0.8 mM Cys-SSH treatment. Conversely, at lower concentrations (0.4 mM and 0.05 mM), there was no statistically significant variance observed compared to the control group (Appendix A). These results suggest that the parasite’s antioxidant system is compromised in the presence of 0.8 mM Cys-SSH.

### 3.4. Comparison of S-Sulfurated and S-Nitrosylated Proteins in E. histolytica Trophozoites

Based on current knowledge, it is conceivable that some reactive cysteine could be modified by either S-sulfuration or S-nitrosylation [59]. A comparative analysis between our study and previous findings [60] reveals that 41 out of 117 S-nitrosylated proteins (fold change > 2) also undergo S-sulfuration in this study. The classification of these 41 S-sulfurated proteins was performed using the PANTHER classification tool (Figure 9a). Eight distinct families of these proteins were identified, including protein modifying enzyme (PC00260) (exemplified by Aminopeptidase [EHI_008380]), transporter (PC00227) (exemplified by Calcium-transporting ATPase [EHI_030830]), membrane traffic protein (PC00150) (exemplified by Coatomer subunit gamma [EHI_040700]), chaperone (PC00072) (70 kDa heat shock protein [EHI_199590]), protein-binding activity modulator (PC00095) (exemplified by Ras family GTPase [EHI_137700]), cytoskeletal protein (PC00085) (exemplified by Actin-related protein 2_3 complex subunit 4 [EHI_030820]), translational protein (PC00263) (exemplified by 60S ribosomal protein L10 [EHI_044810]), and metabolite interconversion enzyme (PC00262) (exemplified by Galactokinase [EHI_094100]). A PANTHER statistical overrepresentation test revealed significant enrichment (*p*-value and q-value < 0.05) for proteins belonging to vesicle coat protein (PC00235) (exemplified by Coatomer subunit gamma EHI_040700), dehydrogenase (PC00092) (exemplified by dihydropyrimidine dehydrogenase (NADP(+)) [EHI_012980]), oxidoreductase (PC00176) (exemplified by Steroid 5-alpha reductase [EHI_076870]), ribosomal protein (PC00202) (exemplified by 60S ribosomal protein L10 [EHI_044810]), membrane traffic protein (PC00150) (exemplified by AP complex subunit beta [EHI_023600]), translational protein (PC00263) (exemplified by 40S ribosomal protein S3 [EHI_146340]), metabolite interconversion enzyme (PC00262) (exemplified by Alcohol dehydrogenase [EHI_088020]), carbohydrate kinase (PC00065) (exemplified by Galactokinase [EHI_094100]), aminoacyl–tRNA synthetase (PC00047) (exemplified by asparagine–tRNA ligase [EHI_126920]), and actin or actin-binding cytoskeletal protein (PC00041) (exemplified by actin-related protein 2_3 complex subunit 4 [EHI_030820]) (Figure 9b).

## 4. Discussion

Apart from the known inhibitory effect of H_2_S on the growth of *Plasmodium falciparum*, resulting in the limitation of cerebral malariasis development in murine models [61,62], our understanding of the impact of RSS on protozoan parasites remains limited. This study investigates the effects of H_2_S and Cys-SSH on *E. histolytica*, shedding light on their cytotoxicity and the involvement of protein S-sulfuration, thereby unraveling potential implications for amebiasis pathogenesis and therapeutic interventions. By employing a BTA in conjunction with LC-MS/MS, we identified 546 S-sulfurated proteins, highlighting the large scope of proteins and pathways potentially affected by this post-translational modification. These findings underscore the far-reaching implications of H_2_S-induced effects on the parasite. The discussion revolves around the potential functional impairment caused by S-sulfuration of multiple proteins and elucidation of how S-sulfuration contributes to the anti-amebic effect of H_2_S.

The Cys-SSH solution may contain some unreacted Na_2_S, cystine, and cysteine trisulfide alongside Cys-SSH [63]. Cystine and Na2S, at concentrations used to prepare Cys-SSH, do not affect the viability of the parasite (this work and [64]). Additionally, they do not independently lead to persulfide formation in the parasite (this work). These observations strongly suggest that Cys-SSH plays a major role in the formation of S-sulfurated proteins and, consequently, in affecting the viability of the parasite. However, we cannot completely rule out that some of the observations attributed to Cys-SSH are caused by cysteine trisulfide.

Many of the S-sulfurated proteins in the parasite are related to redox regulation, including peroxiredoxin and thioredoxin. Peroxiredoxin serves as the primary thiol-containing surface antigen of *E. histolytica* and it is involved in the resistance of the parasite to oxidative stress by scavenging reactive oxygen species [65]. *E. histolytica* thioredoxin, thioredoxin reductase together with peroxiredoxin mediate NADPH-dependent reduction of H_2_O_2_ [66]. Protein S-sulfuration and S-nitrosylation exhibit numerous similarities concerning their chemical and biological characteristics [59]. The identification of peroxiredoxin within the subset of S-sulfurated and S-nitrosylated proteins in the parasite subjected to nitrosative stress is consistent with existing reports indicating that thiols susceptible to S-nitrosylation are often targets for sulfhydration as well [67]. While our study did not specifically investigate the impact of S-sulfuration on the activity of these antioxidant enzymes, the observed accumulation of ROS in the parasite following exposure to Cys-SSH strongly suggests that elevated persulfidation disrupts the parasite’s antioxidant defense mechanisms. Given the critical role of antioxidant enzymes for the parasite’s survival, any impairments via their S-sulfuration can be detrimental to the parasite’s viability. At the same time, S-sulfuration appears to interfere with the parasite’s antioxidant defenses in various other biological systems, such as the cardiovascular and renal systems. H_2_S functions by deactivating oxidant species and amplifying the expression of internal antioxidant defenses [68]. These apparently opposite effects can be attributed to the concentration of H_2_S, which can be toxic at mM concentrations, as observed in our study, but protective at micromolar ranges [69].

Heat shock proteins (HSPs) play a fundamental role in safeguarding the parasite against various environmental stresses, including temperature fluctuations and host immune responses. HSPs are involved in protein folding, assembly, and degradation, ensuring proper cellular function under stress conditions, including ER stress, oxidative stress, and nutrient deprivation [70,71,72]. These proteins are also important during the differentiation of the parasite to cysts [73]. HSPs 70 (EHI_001950, EHI_199590) and HSP90 (EHI_163480) were identified among S-sulfurated proteins present in the parasite. The presence of HSPs among S-sulfurated proteins in *E. histolytica* is in agreement with the prominent detection of HSP70 as a major S-sulfurated protein in another eukaryotic system [74]. However, the precise impact of S-sulfuration on the chaperone activity of HSPs remains unclear. In mammalian HSP90, the oxidation of reactive cysteine residues was shown to impair chaperone activity [75]. Consequently, it is plausible that S-sulfuration of essential cysteine residues may also modulate the activity of HSPs in *E. histolytica*. The discovery of HSPs among sulfhydrated proteins in this parasite raises intriguing questions about the role of this modification in regulating the parasite’s response to stresses and its differentiation. Drawing from the example of HSP90 inhibitors that effectively impede parasite growth at the micromolar range, underscoring the critical role of HSPs for the parasite [76], future investigations could explore the potential impact of S-sulfuration on HSP activity.

Both HSPs and protein synthesis are linked to environmental stress. HSPs actively contribute to a protective response against cellular stress [77]. Moreover, inhibiting protein synthesis redirects energy resources that would typically be utilized for synthesizing proteins, channeling it instead toward repairing stress-induced cellular damage [78,79]. Our previous studies showed that oxidative stress and nitrosative stress inhibit protein synthesis in *E. histolytica* [38,55]. Here, we observed that Cys-SSH inhibits protein synthesis in the parasite. We identified many proteins related to translation that undergo sulfhydration, including eIF2α [EHI_005100], Serine/Threonine-Protein Phosphatase [EHI_117570], Protein Phosphatase [EHI_110320, EHI_086040], Elongation Factor 2 [EHI_189490], and ribosomal proteins. Similar to their vulnerability to oxidation and S-nitrosylation [80,81], ribosomal proteins contain cysteine residues, which could potentially be targets for S-sulfuration. Alterations in these residues due to S-sulfuration might affect ribosome assembly, stability, or interaction with other molecules involved in the translation process, leading to the inhibition of protein synthesis. In mammalian cells, H_2_S transiently enhances the phosphorylation of eIF2α, leading to a general inhibition of protein synthesis [82]. Given that various stressors such as prolonged serum starvation, heat shock, and oxidative stress induce an elevation in phospho-EheIF2α levels within the parasite [83], it is plausible that the S-sulfuration of proteins could also contribute to the increased levels of phospho-EheIF2α and, consequently, to an inhibition of protein synthesis.

*E. histolytica* relies on actin and its polymerized form, F-actin, for various biological processes crucial for its survival and virulence, including cell movement, phagocytosis, adherence to host cells, and tissue invasion. In previous studies, we demonstrated that S-oxidation and S-nitrosylation of actin lead to the inhibition of its polymerization [38,55]. The identification of cytoskeletal proteins such as actin and tubulin as S-sulfurated proteins in the parasite indicates a potential regulatory role of this post-translational modification in cytoskeletal function within this organism. This hypothesis gains support from our observations of reduced migration activity in the parasite when exposed to Cys-SSH, with this diminished activity being dependent on actin and reduced F-actin formation. In *Arabidopsis thaliana*, the accumulation of H_2_S causes the sulfhydration of a crucial cysteine residue at Cys-287 in ACTIN2, leading to the destabilization of F-actin filaments [84]. Intriguingly, this specific residue is also found in *E. histolytica* actin, which shares an 87% identity with *A. thaliana* actin, implying that a similar inhibitory mechanism might apply to this parasite as well.

EhCPs are key virulence factors of *E. histolytica* and function in intestinal invasion by degrading the extracellular matrix and avoiding the host immune response by breaking down secretory immunoglobulin A and immunoglobulin G and triggering the complement system [85]. We recently reported that *E. histolytica* trophozoites exhibit the ability to degrade bacterial cell aggregates, known as biofilms, through their CP activity [45]. In this study, EhCP1 [EHI_074180], EhCP4 [EHI_050570], EhCP-C3 [EHI_138460], EhCP-C8 [EHI_182770], and EhCP [EHI_050800] were found among the S-sulfurated proteins. The S-nitrosylation of the cysteine catalytic residue of CPs may lead to inhibition of its activity in *Plasmodium falciparum* trophozoites and *E. histolytica* trophozoites [86,87], resulting in a 50% reduction in damage to tissue culture monolayer cells in the case of *E. histolytica* [86]. Our study shows that S-sulfuration leads to a significant reduction in the activity of EhCPs, which can be the result of the modification of its cysteine catalytic residues. As a result, the capacity of Cys-SSH-treated *E. histolytica* trophozoites to degrade *B. subtilis* biofilms was diminished. These discoveries underscore the intricate relationship among S-sulfuration, EhCPs activity, and biofilm degradation. Moreover, the production of H_2_S by the gut microbiota might directly impact crucial functions within the parasite, such as its ability to degrade bacterial biofilms. Despite the existence of numerous well-established medicines containing sulfur moieties that may release H_2_S in vivo, the prospect of utilizing H_2_S as a therapeutic agent is still in its infancy [88]. Interestingly, natural products like garlic are capable of releasing H_2_S when their derived sulfur compounds are in contact with reduced glutathione [89]. While allicin is the main anti-amebic compound in garlic [90], the specific mechanisms and extent of H_2_S release from garlic’s sulfur compounds remain areas requiring further investigation to fully understand its potential as an anti-amebic agent.

## 5. Conclusions

In conclusion, this study provides the first comprehensive insights into the intricate interplay between RSS and *E. histolytica*, elucidating their cytotoxic effects, proteomic alterations, and subsequent functional consequences. These findings pave the way for further research exploring the specific pathways affected by S-sulfuration and its broader implications for amebiasis pathogenesis and therapeutic strategies.

## Figures and Tables

**Figure 1 antioxidants-13-00245-f001:**
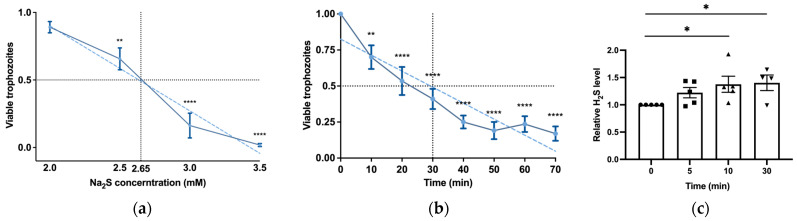
H_2_S induces rapid cytotoxicity and intracellular accumulation in *E. histolytica* trophozoites. (**a**) The viability of Na_2_S-treated E. *histolytica* trophozoites using different concentrations. The 1 × 10^6^ trophozoites were exposed to different Na_2_S concentrations for 30 min in PBS–glucose. The obtained data were normalized and compared to untreated trophozoites from biological experiments conducted on three separate days, each with one to three independent repetitions. Data represent the mean ± SEM of six independent replicates (*n* = 6). (**b**) The viability of Na_2_S-treated *E. histolytica* trophozoites was assessed over various time intervals. Specifically, 1 × 10^6^ trophozoites were subjected to Na_2_S at a concentration of 2.65 mM in PBS–glucose for different durations to evaluate the time-dependent effects on viability. The obtained data were normalized and compared to untreated trophozoites from biological experiments conducted on three separate days, each with two to three independent repetitions. Data represent the mean ± SEM of seven independent replicates (*n* = 7). (**c**) Detection of H_2_S production by flow cytometry. Exposure of *E. histolytica* to 2.65 mM Na_2_S at various time points resulted in the generation of H_2_S. The H_2_S accumulation was assessed using the WSP-5 fluorescent probe. The obtained data were normalized and compared to untreated trophozoites from biological experiments conducted on two separate days, each with two to three independent repetitions. Data represent the mean ± SEM of five independent replicates (*n* = 5). Statistical significance is indicated by asterisks (* *p* < 0.05, ** *p* < 0.01, **** *p* < 0.0001).

**Figure 2 antioxidants-13-00245-f002:**
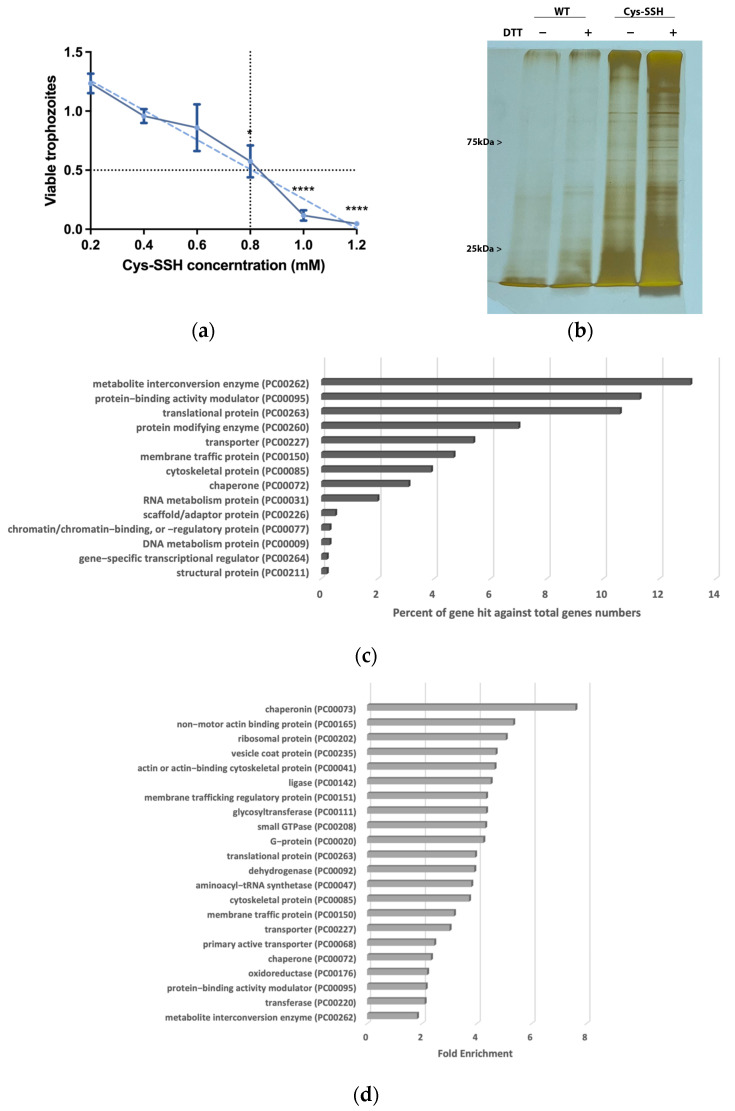
Proteomic profiling of S-sulfurated proteins in parasites exposed to Cys-SSH by BTA reveals diverse functional families. (**a**) Effect of Cys-SSH on viability of E. *histolytica* trophozoites. The 1 × 10^6^ trophozoites were exposed to different concentrations of Cys-SSH for 30 min in PBS–glucose. The obtained data were normalized and compared to untreated trophozoites from biological experiments conducted on three separate days, each with one to two independent repetitions. Data represent the mean ± SEM of four independent replicates (*n* = 4). Statistical significance was indicated by asterisks (* *p* < 0.05, **** *p* < 0.0001). (**b**) Analysis of protein persulfidation in *E. histolytica* treated with and without Cys-SSH by BTA. Silver staining revealed that trophozoites treated with 0.8 mM Cys-SSH for 30 min produced more S-sulfurated protein compared to control trophozoites (WT). DTT, dithiothreitol. (**c**) PANTHER classification of the 546 S-sulfurated proteins identified in Cys-SSH-treated *E. histolytica* trophozoites. (**d**) The PANTHER statistical overrepresentation analysis was performed on the 546 S-sulfurated proteins identified in *E. histolytica* trophozoites treated with Cys-SSHs.

**Figure 3 antioxidants-13-00245-f003:**
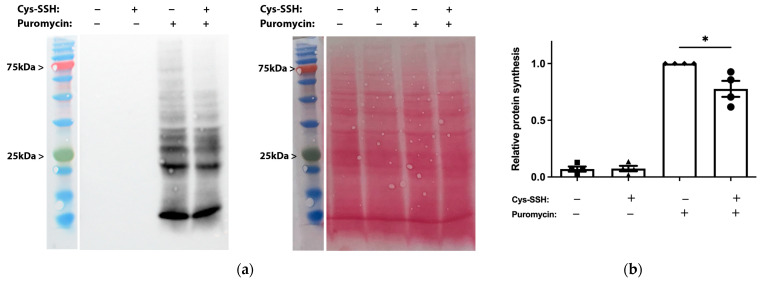
Determination by the SUnSET of protein synthesis in Cys-SSH-treated trophozoites. (**a**) Left: protein synthesis was assessed using Western blot with puromycin antibody. The 2 × 10^6^ trophozoites were treated with or without 0.8 mM Cys-SSH for 30 min and then labelled with or without 10 μg/mL puromycin for 20 min. The protein extracts underwent separation via denaturing electrophoresis and were subsequently examined through immunoblotting using a monoclonal puromycin antibody. Right: ponceau S stain total protein labeling. (**b**) Normalization using Fiji software version 1.54f of the puromycin signal relative to the total protein signal in control trophozoites. The normalized values for control trophozoites labeled with puromycin were set to 1. The obtained data were normalized and compared to untreated trophozoites from biological experiments conducted on two separate days, each with two independent repetitions. Data represent the mean ± SEM of four independent replicates (*n* = 4). Statistical significance is denoted by asterisks (* *p* < 0.05).

**Figure 4 antioxidants-13-00245-f004:**
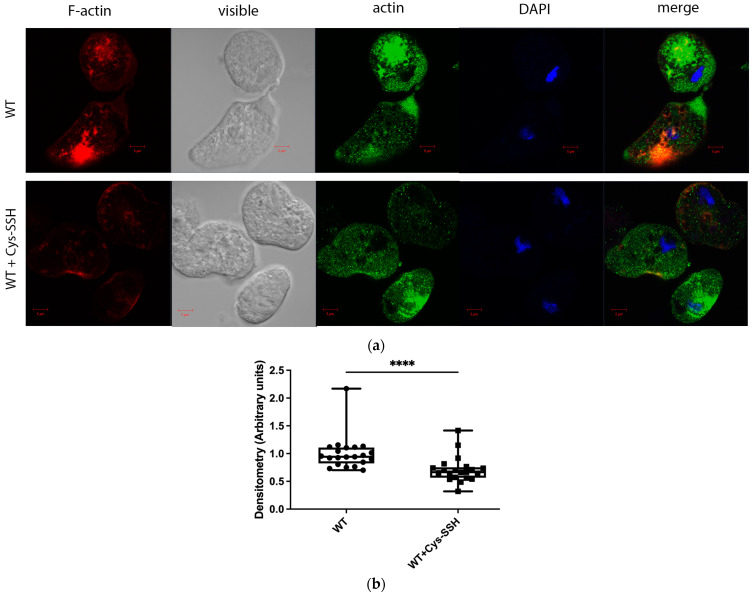
The level of F-actin was reduced in Cys-SSH-treated *E. histolytica* trophozoites. (**a**) Confocal laser scanning microscopy employing a 63× oil immersion lens objective was utilized to observe F-actin and total actin in both untreated *E. histolytica* trophozoites and those treated with Cys-SSH. F-actin (red) was visualized using Phalloidin-iFluor 594 Reagent, while total actin (green) was detected using a primary actin antibody followed by a secondary Alexa Flour 488 antibody. Nuclei were stained blue using DAPI dye. (**b**) Evaluation of the F-actin fluorescence ratio to total actin fluorescence in control trophozoites and those treated with Cys-SSH. The analysis involved 21 trophozoites and was conducted using the Fiji software version 1.54f, with the signal from control trophozoites set as the reference value of 1. **** *p* < 0.0001.

**Figure 5 antioxidants-13-00245-f005:**
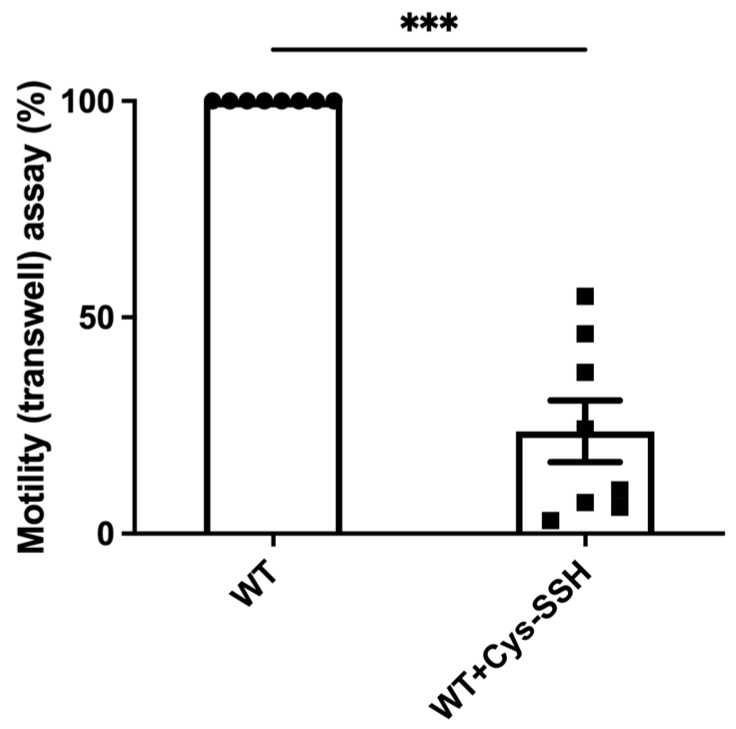
The motility of *E. histolytica* trophozoites was impaired by Cys-SSH. The motility of control and *E. histolytica* trophozoites treated with Cys-SSH (0.8 mM, 30 min) were examined. The corresponding value of control trophozoites, which were not subjected to Cys-SSH, was designated as 100%. The obtained data were normalized and compared to untreated trophozoites from biological experiments conducted on two separate days, each with four independent repetitions. Data represent the mean ± SEM of eight independent replicates (*n* = 8). Statistical significance was denoted by asterisks (*** *p* < 0.001).

**Figure 6 antioxidants-13-00245-f006:**
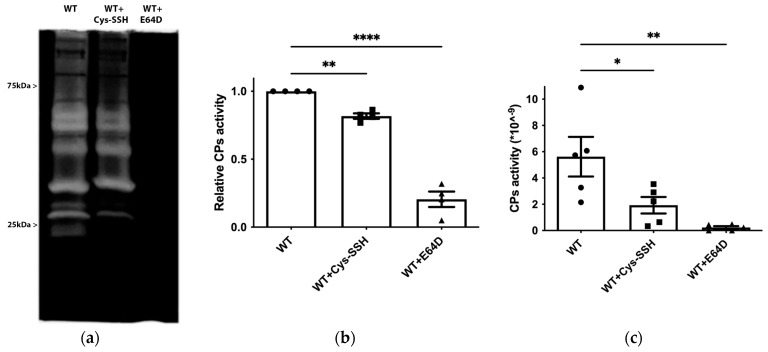
Inhibition of CP activity by Cys-SSH in E. *histolytica.* (**a**) Detection of CP activity using gelatin gel (1% gelatin in 12% SDS gel, followed by staining with Coomassie) in control trophozoites (WT), Cys-SSH-treated trophozoites (0.8 mM Cys-SSH for 30 min) and in E64D-treated trophozoites (10 µM E64D for overnight). (**b**) Densitometry analysis of gelatin gel showing relative CPs activity of different conditions trophozoites. The obtained data were normalized and compared to untreated trophozoites from biological experiments conducted on two separate days, each with two independent repetitions. Data represent the mean ± SEM of four independent replicates (*n* = 4). (**c**) CP activity was assessed by tracking the cleavage of the substrate z-Arg-Arg-pNA monitored by a spectrophotometric method in WT, Cys-SSH-treated trophozoites, and E64D-treated trophozoites. The obtained data were normalized and compared to untreated trophozoites from biological experiments conducted on three separate days, each with one to two independent repetitions. Data represent the mean ± SEM of five independent replicates *(n* = 5). Statistical significance is indicated by asterisks (* *p* < 0.05, ** *p* < 0.01, **** *p* < 0.0001).

**Figure 7 antioxidants-13-00245-f007:**
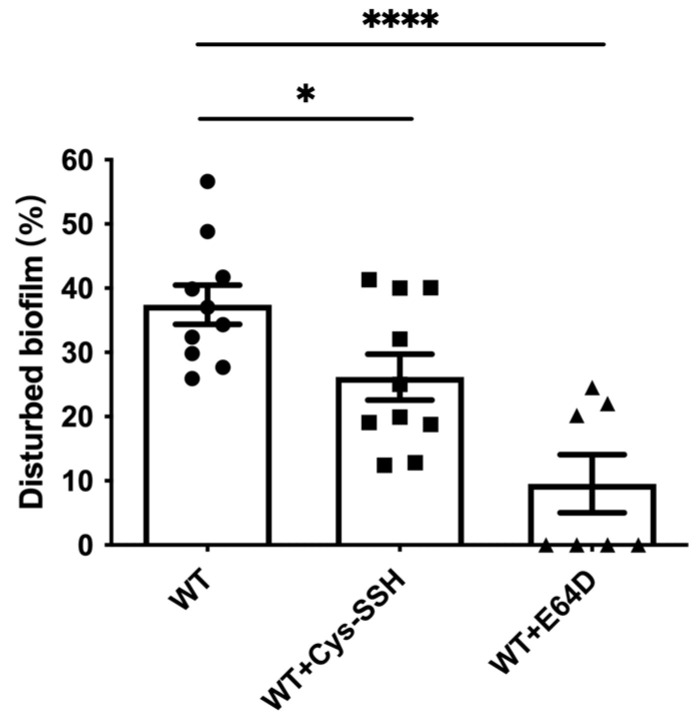
Degradation of *B. subtilis* biofilm by *E. histolytica* trophozoites treated with Cys-SSH. The breakdown of *B. subtilis* biofilm was evaluated following a 3 h incubation period with control trophozoites (WT) and trophozoites exposed to 0.8 mM Cys-SSH for 30 min (WT + Cys-SSH) and trophozoites treated with 10 μM E64D for overnight (WT + E64D). The obtained data were normalized and compared to untreated trophozoites from biological experiments conducted on three separate days, each with three to four independent repetitions. Data represent the mean ± SEM of ten independent replicates (*n* = 10). Statistical significance is denoted by asterisks (* *p* < 0.05, **** *p* < 0.0001).

**Figure 8 antioxidants-13-00245-f008:**
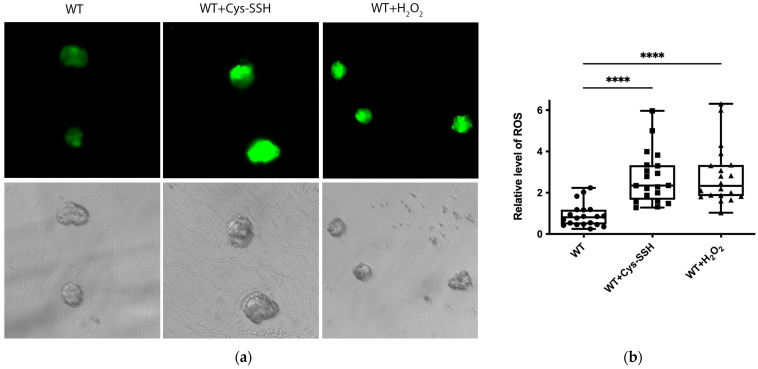
The ROS level of *E. histolytica* trophozoites treated with Cys-SSH. (**a**) Fluorescence microscopy (20×) of *E. histolytica* trophozoites, trophozoites treated with 0.8 mM Cys-SSH for 30 min, and trophozoites subjected to 2.5 mM H_2_O_2_ for 15 min. The trophozoites were labeled with the probe H2DCFDA, leading to a green fluorescence indicative of the presence and quantities of ROS within each trophozoite. (**b**) The relative fluorescence intensity level of H2DCFDA was compared between control trophozoites, Cys-SSH-treated trophozoites, and H_2_O_2_-treated trophozoites. The examination was conducted using the Fiji software version 1.54f on a sample of 20 trophozoites. The signal from control trophozoites was established as 1. **** *p* < 0.0001.

**Figure 9 antioxidants-13-00245-f009:**
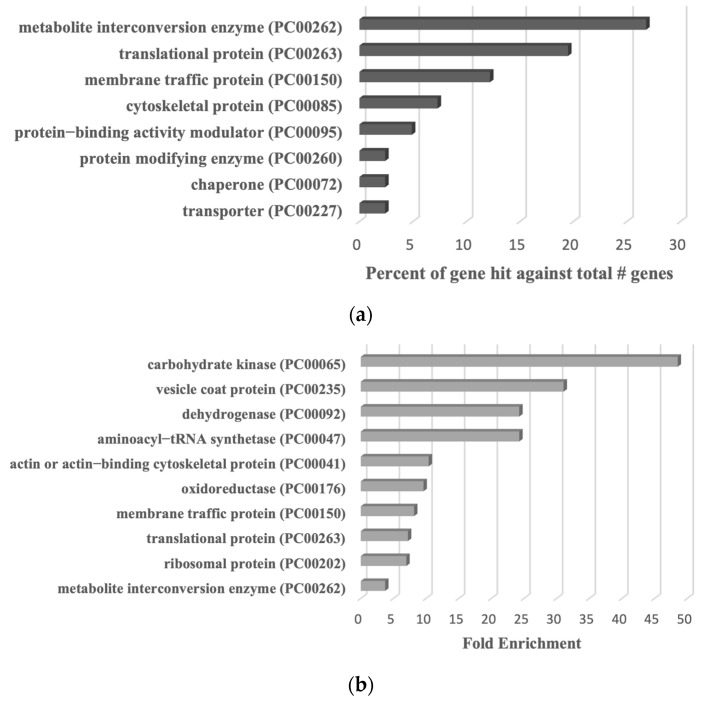
PANTHER analysis of common proteins undergoing S-sulfuration and S-nitrosylation in *E. histolytica* trophozoites. (**a**) PANTHER classification of the 41 common proteins undergoing S-sulfuration and S-nitrosylation. (**b**) PANTHER statistical overrepresentation test of the 41 common proteins undergoing S-sulfuration and S-nitrosylation.

## Data Availability

Data are contained within the article and Appendix A.

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
