# Peer review of "Impact of Reactive Sulfur Species on Entamoeba histolytica: Modulating Viability, Motility, and Biofilm Degradation Capacity"

_antioxidants, 2024, doi:10.3390/antiox13020245_

Round 1
Reviewer 1 Report
Comments and Suggestions for Authors
The authors explored the effects of protein persulfidation on vital parameters and cellular functions of throphozoites of E. histolytica, a gut parasite. Overall, this is an interesting and well-performed study; the manuscript is well-written and fits into the scope of the journal. Nevertheless, I have some questions/remarks regarding the experimental design of the study and the conclusions:
1) Why only the first set of experiments was done with NaHS, whereas all subsequent experiments were done with Cys-SSH? This design is not properly reflected in the title and in the conclusions. It is also somewhat problematic with respect to potential pathophysiological implications: As compared to NaHS, Cys-SSH was more toxic to the trophozoites, and some of the here observed effects of Cys-SSH may not occur, when the trophozoites are exposed to NaHS or a slow-releasing H2S donor. This is even more probable, as trophozoites residing in the colon in vivo appear to tolerate well the millimolar concentrations of H2S that commonly occur in the colonic lumen. This point should be discussed as a limitation of the study.
2) To my mind, another limitation of the study should be discussed with respect to the therapeutic implications suggested by the authors. H2S is well known to exert toxic effects at higher doses. Thus, pronounced protein persulfidation is likely to impair not only the viability of throphozoites (as shown here) but also to induce cytotoxicity in the human host, e.g. through disturbing the respiratory chain in the colonocytes and perhaps even impair the integrity of the colonic epithelium.
Minor points:
1) Introduction, Lanes 35-37: Humans possess a fourth endogenous H2S-producing enzyme, which is particularly important in the colon: SELENBP1 produces H2S through oxidation of methanethiol, a product of the methionine metabolism of gut bacteria.
2) Introduction, Lanes 46-48: Please try to rephrase these very redundant statements.
Author Response
February 12, 2024
Dear Reviewers of our manuscript antioxidants-2842388,
We would like to thank you for your constructive comments. Below, you will find point-by-point responses addressing each of your remarks. An updated version of the manuscript has been uploaded, with all revisions highlighted in red fonts. We trust that this revised edition sufficiently addresses the concerns you have raised.
Thank you once again for your valuable input.
Best regards
Serge Ankri on behalf of the authors.
Reviewer 1 Answers:
Point 1: Why only the first set of experiments was done with NaHS, whereas all subsequent experiments were done with Cys-SSH? This design is not properly reflected in the title and in the conclusions. It is also somewhat problematic with respect to potential pathophysiological implications: As compared to NaHS, Cys-SSH was more toxic to the trophozoites, and some of the here observed effects of Cys-SSH may not occur, when the trophozoites are exposed to NaHS or a slow-releasing H2S donor. This is even more probable, as trophozoites residing in the colon in vivo appear to tolerate well the millimolar concentrations of H2S that commonly occur in the colonic lumen. This point should be discussed as a limitation of the study.
Answer:
Na2S, a well-established H2S donor, served as a starting point to establish the effects of H2S on trophozoites under our experimental conditions. Na2S, as a donor, indeed does not lead directly to S-sulfhydration unless cysteines are oxidized. Given that the trophozoites thrive in the colon, where the environment is predominantly anaerobic unless inflammation occurs, it is reasonable to assume that cysteines within the parasite's proteins are predominantly in their reduced state. To directly study S-sulfhydration under conditions reflective of the colonic environment, we chose Cys-SSH for this purpose. Importantly, Cys-SSH is naturally present in the gut. We have incorporated a new reference in the manuscript to support this information.
The title and conclusions have been revised, and the article now addresses the "Impact of reactive sulfur species" instead of H2S.
We agree with reviewer #1 regarding the necessity of addressing the heightened toxicity of Cys-SSH compared to Na2S. The updated manuscript now highlights that Cys-SSH can directly induce S-sulfhydration of cysteine residues in essential parasite proteins, including cysteine proteases and others discussed in this study. In contrast, the sulfhydration of cysteine residues by Na2S requires prior oxidation of the cysteine residues. It is worth noting that such oxidation is less prone to occur in the colon, which is characterized by a reduced environment.
Point 2: To my mind, another limitation of the study should be discussed with respect to the therapeutic implications suggested by the authors. H2S is well known to exert toxic effects at higher doses. Thus, pronounced protein persulfidation is likely to impair not only the viability of throphozoites (as shown here) but also to induce cytotoxicity in the human host, e.g. through disturbing the respiratory chain in the colonocytes and perhaps even impair the integrity of the colonic epithelium.
Answer:
In contrast to the parasite, the human host can metabolize H2S through its oxidation by the mitochondrial sulfide oxidation pathway. It has also been recently demonstrated that cysteine persulfide (Cys-SSH) can be reduced in the mitochondria by accepting an electron. Therefore, unlike the parasite, mammalian cells possess mechanisms to handle H2S and Cys-SSH that the parasite lacks. This information has been added to the discussion. doi: 10.1038/s41467-017-01311-y doi: 10.1186/s12986-021-00598-5
Minor points:
1) Introduction, Lanes 35-37: Humans possess a fourth endogenous H2S-producing enzyme, which is particularly important in the colon: SELENBP1 produces H2S through oxidation of methanethiol, a product of the methionine metabolism of gut bacteria.
2) Introduction, Lanes 46-48: Please try to rephrase these very redundant statements.
Answer:
The minors point have been addressed in the revised version of the manuscript

Reviewer 2 Report
Comments and Suggestions for Authors
The study conducted by Ye et al. aims to elucidate the role of hydrogen sulfide (H2S) in the protozoan parasite Entamoeba histolytica. The experiment began by investigating the effects of H2S on E. histolytica, revealing that exposure to Na2S resulted in rapid cytotoxicity, with an LD50 of 2.65 mM. Additionally, the persulfidating agent cysteine-persulfide (Cys-SSH) exhibited cytotoxic effects, showing an LD50 of 0.8 mM. The researchers conducted a proteomic analysis, identifying over 500 S-sulfhydrated proteins in trophozoites challenged with Cys-SSH. Functional assessments demonstrated that Cys-SSH inhibits protein synthesis, alters cytoskeletal dynamics, and reduces motility in trophozoites. Moreover, the study revealed that Cys-SSH inhibits cysteine proteases (CPs) and impacts bacterial biofilm degradation capacity. While the study presents valuable insights into H2S regulation in the parasite and is well-presented, some concerns need to be addressed before considering the manuscript for publication.
Comments-
1. The authors employed a method involving the reaction of cystine with sodium sulfide to generate Cys-SSH. However, this approach presents challenges in Cys-SSH synthesis due to the dynamic equilibrium among cystine, H2S, Cys-SSH, Cys-SH, and cysteine trisulfide (British Journal of Pharmacology, 2019, 176, 671–683). Evaluating the exclusive effects of Cys-SSH becomes challenging under these circumstances. Additionally, the presence of cysteine trisulfide, known to generate signals in cold analysis methods, introduces a potential interference in accurately estimating Cys-SSH concentrations. In light of these limitations, it would be insightful for the authors to elaborate on the rationale behind choosing this method for Cys-SSH production.
2. In the biotin thiol assay, the authors employed biotin-conjugated maleimide to label persulfides and thiols. However, recent studies indicate that maleimide-based reagents may be unsuitable for preserving persulfides under biologically relevant conditions. The resulting persulfide adducts (R-S-S-maleimide) convert into corresponding thioethers (R-S-maleimide) (Angew. Chem. Int. Ed. 2022, 61, e202203684). Additionally, maleimide-based labeling agents have the potential to disrupt polysulfide chains, leading to the formation of NEM-derivatized polysulfide adducts (Br J Pharmacol 2019,176, 646–670). Furthermore, in the reduction step, DTT can also reduce disulfide bonds within peptides. Given these limitations, it raises the question of why the authors opted for a maleimide-based reagent in their study.
3. A rise in ROS levels in trophozoites treated with Cys-SSH was observed. The authors proposed that this may be linked to the concentration of H2S, demonstrating toxicity at mM concentrations but a protective role at micromolar ranges. A query arises: Have the authors conducted measurements of ROS levels at micromolar concentrations of Cys-SSH?
4. The manuscript's title requires modification. Presently, it is titled "Impact of hydrogen sulfide on Entamoeba histolytica:." However, since the majority of the studies employed cysteine hydropersulfide (Cys-SSH). As clarified in the manuscript, Cys-SSH acts as persulfidating agent rather than an H2S donor, this distinction should be incorporated into the title.
Author Response
February 12, 2024
Dear Reviewers of our manuscript antioxidants-2842388,
We would like to thank you for your constructive comments. Below, you will find point-by-point responses addressing each of your remarks. An updated version of the manuscript has been uploaded, with all revisions highlighted in red fonts. We trust that this revised edition sufficiently addresses the concerns you have raised.
Thank you once again for your valuable input.
Best regards
Serge Ankri on behalf of the authors.
Reviewer 2 Answers:
Point 1: The authors employed a method involving the reaction of cystine with sodium sulfide to generate Cys-SSH. However, this approach presents challenges in Cys-SSH synthesis due to the dynamic equilibrium among cystine, H2S, Cys-SSH, Cys-SH, and cysteine trisulfide (British Journal of Pharmacology, 2019, 176, 671–683). Evaluating the exclusive effects of Cys-SSH becomes challenging under these circumstances. Additionally, the presence of cysteine trisulfide, known to generate signals in cold analysis methods, introduces a potential interference in accurately estimating Cys-SSH concentrations. In light of these limitations, it would be insightful for the authors to elaborate on the rationale behind choosing this method for Cys-SSH production.
Answer:
The method we employed to produce Cys-SSH is a commonly used approach, as documented in previous publications. Consequently, we did not consider the issues raised by reviewer #2. Cys-SSH is generated by reacting cystine with Na2S. At the concentrations utilized for Cys-SSH preparation, cystine enhances the parasite's growth, as evidenced by the reference https://doi.org/10.1016/0014-4894(81)90125-9, while Na2S is non-toxic. Additionally, we present new experimental evidence demonstrating that cystine or Na2S alone are incapable of triggering the formation of persulfide species, as detected by the SSP-4 probe. In contrast, Cys-SSH, formed through the reaction of cystine with Na2S, activates the SSP-4 probe in the exposed parasite. This information added to the revised version of the manuscript strongly suggest that the anti-amebic effect is indeed due to the formation of persulfide. We acknowledge that distinguishing between the effects of Cys-SSH and cysteine trisulfide is challenging, as both are detected by the SSP-4 probes. The limitations of this approach are now addressed in the discussion.
Point 2. In the biotin thiol assay, the authors employed biotin-conjugated maleimide to label persulfides and thiols. However, recent studies indicate that maleimide-based reagents may be unsuitable for preserving persulfides under biologically relevant conditions. The resulting persulfide adducts (R-S-S-maleimide) convert into corresponding thioethers (R-S-maleimide) (Angew. Chem. Int. Ed. 2022, 61, e202203684). Additionally, maleimide-based labeling agents have the potential to disrupt polysulfide chains, leading to the formation of NEM-derivatized polysulfide adducts (Br J Pharmacol 2019,176, 646–670). Furthermore, in the reduction step, DTT can also reduce disulfide bonds within peptides. Given these limitations, it raises the question of why the authors opted for a maleimide-based reagent in their study.
Answer:
It is important to note that the concentration of NEM used in this study is 10 times lower than the concentration reported in the article referenced by reviewer #2. Therefore, some of the side effects mentioned by the reviewer are likely mitigated due to the lower concentration employed in our study. Regarding the reduction step with DTT, it is crucial to highlight that we are performing the BTA assay on proteins, not peptides, as reported in the reference provided by the reviewer. This distinction is significant because it limits the reduction of disulfide bonds within peptides. Even if all the side effects described by the reviewer were to occur, we would expect to observe fewer S-sulfhydrated proteins than what we have identified in this study. However, it is noteworthy that we have successfully identified more than 500 S-sulfhydrated proteins, which is more than sufficient for drawing meaningful conclusions from our investigation. In addition, recent papers, have successfully used BTA, such as PMID 33741971.
Point 3. A rise in ROS levels in trophozoites treated with Cys-SSH was observed. The authors proposed that this may be linked to the concentration of H2S, demonstrating toxicity at mM concentrations but a protective role at micromolar ranges. A query arises: Have the authors conducted measurements of ROS levels at micromolar concentrations of Cys-SSH?
Answer:
In the revised version of the manuscript, we examined the level of ROS in the parasite exposed to various concentrations of Cys-SSH, including those in the micromolar range. Our observations revealed that the ROS levels in the parasite exposed to micromolar concentrations of Cys-SSH is not significantly increased compared to the untreated parasite.
Point 4. The manuscript's title requires modification. Presently, it is titled "Impact of hydrogen sulfide on Entamoeba histolytica:." However, since the majority of the studies employed cysteine hydropersulfide (Cys-SSH). As clarified in the manuscript, Cys-SSH acts as persulfidating agent rather than an H2S donor, this distinction should be incorporated into the title.
Answer:
The title has been revised in accordance with the recommendations made by the reviewer.
Reviewer 3 Report
Comments and Suggestions for Authors
In the present study authors examined the effect of hydrogen sulfide on Entameba histolytica biology. The study was inspired by the presence of high amounts of H2S produced by microbiota in the gastrointestinal tract. E. histolytica trophozites were treated with either Na2S or cysteine persulfide and proteomic analysis of parasite proteins was performed regarding their sulfuration. The results show that both Na2S and Cys-SSH reduce parasite viability which was associated with the increase in intracellular H2S concentration and sulfuration of 546 proteins. The function of modified protein families was identified by bioinformatics analysis. Cys-SSH inhibited protein translation as demonstrated by SUnSET method. Cys-SSH decreased the level of F-actin formed in trophozoites. Consistently with this finding, CysSSH reduced motility of trophozoites in transwell migration assay. In addition, Cys-SSH reduced the activity of E. histolytica cysteine proteases important for its virulence. Cys-SSH treated trophozoites were impaired in their ability to degrade B. subtilis biofilm compared to the untreated trophozoites. Interestingly, a significant increase in ROS levels in trophozoites treated with Cys-SSH, comparable to the ROS levels observed after H2O2 treatment, was demonstrated.
The topic is of interest and the results contribute much to our understanding of the role of H2S in the gastrointestinal tract. I have some comments how the manuscript could be improved.
1) Section 2.2: it is well known that Na2S is spontaneously oxidized in solution to form other reactive sulfur species with some overlapping but others divergent effects than H2S. It should be described how Na2S solution was prepared and handled to minimize the risk of such transformation.
2) What attempts have been made to ensure that prepared Cys-SSH solution was free of contamination with reagents used for its preparation, in particular Na2S itself?
3) “S-sulfhydration” should be corrected to “S-sulfuration” which is the more appropriate name since “sulfhydration” suggests adding sulfhydryl group whereas the modification is conversion of existing –SH groups to –SSH groups, that is adding one sulfur atom.
4) Statistical analysis: were all data normally distributed to justify using Student t-test? Was normality of data distribution verified?
5) In some sets of experiments more than 2 groups were compared so ANOVA rather than Student t-test would be more appropriate.
6) Section 3.1.2: the principle of the assay (first paragraph) should be described in the Methods rather than in the Results.
7) Clinical implications of the findings should be discussed. In particular, do the authors believe that H2S present in the GI tract at physiological concentration exerts similar effect on E. histolytica in the course of infection and/or is it possible to reduce E. histolytica viability and manage the infection by pharmacologically increasing H2S level?
Author Response
Dear Reviewers of our manuscript antioxidants-2842388,
We would like to thank you for your constructive comments. Below, you will find point-by-point responses addressing each of your remarks. An updated version of the manuscript has been uploaded, with all revisions highlighted in red fonts. We trust that this revised edition sufficiently addresses the concerns you have raised.
Thank you once again for your valuable input.
Best regards
Serge Ankri on behalf of the authors.
Reviewer 3 Answers:
Point 1. Section 2.2: it is well known that Na2S is spontaneously oxidized in solution to form other reactive sulfur species with some overlapping but others divergent effects than H2S. It should be described how Na2S solution was prepared and handled to minimize the risk of such transformation.
Answer:
Fresh Na2S solution was systematically prepared before each experiment to avoid oxidation. This information is now mentioned in the material and methods section.
Point 2. What attempts have been made to ensure that prepared Cys-SSH solution was free of contamination with reagents used for its preparation, in particular Na2S itself?
Answer:
Kindly refer to our comprehensive response to reviewer 2, addressing the query raised in point 1, which pertains to the same comment.
Point 3. “S-sulfhydration” should be corrected to “S-sulfuration” which is the more appropriate name since “sulfhydration” suggests adding sulfhydryl group whereas the modification is conversion of existing –SH groups to –SSH groups, that is adding one sulfur atom.
Answer: S-sulfhydration is also referred to in the literature as S-sulfuration or S-persulfidation. While we have chosen to maintain the term S-sulfhydration throughout the text, in the Introduction, we have included S-sulfuration (and S-persulfidation) in parentheses the first time this term is mentioned.
Point 4&5 about the statistical analysis
Answer: We have implemented the requested modifications in the statistical analysis tests to analyze our data, including the ANOVA test when applicable. Overall, the conclusions of this study remain unchanged.
Point 6 Section 3.1.2: the principle of the assay (first paragraph) should be described in the Methods rather than in the Results.
Answer: done.
Point 7 Clinical implications of the findings should be discussed. In particular, do the authors believe that H2S present in the GI tract at physiological concentration exerts similar effect on E. histolytica in the course of infection and/or is it possible to reduce E. histolytica viability and manage the infection by pharmacologically increasing H2S level?
Answer: We included a paragraph in the discussion to explore the potential modulation of H2S concentration in the gut and its potential influence on the parasite's development. This consideration is crucial given the significance of maintaining a proper equilibrium, especially since elevated levels of H2S may impact colonic cells.
Round 2
Reviewer 3 Report
The manuscript has been improved and most concerns raised by the reviewers have been adequately addressed by the authors. However, I still have some comments.
1) I strongly suggest to change "S-sulfhydration" to "S-sulfuration" or "persulfidation" throughout the text which are currently the widely acceptable nomenclature. The name "S-sulfhydration" has been widely used in the past but currently it is quite commonly replaced by sulfuration or persufidation.
2) Molar concentration of cystine is much higher than of Na2S in the solution used to prepare CysSSH thus some cystine must remain in the solution. Was it anyhow removed? Otherwise, the effects could be attributed to both CysSSH and cystine
The manuscript has been revised according to the reviewers' comments.
Author Response
Dear Reviewer #3,
1- We have edited the manuscript to replace S-sulfhydration" to "S-sulfuration and S-sulfhydrated" to "S-sulfurated everywhere in the manuscript.
2- As mentioned in our answers to the reviewers, cystine is not toxic to the parasite and even promotes its growth at the concentration used (see reference [66]).
Best regards
Serge Ankri (in behalf of the authors)